# A Machine Learning Model to Reframe the Concept of Shelf-Life in Bakery Products: PDO Sourdough as a Technological Preservation Model

**DOI:** 10.3390/foods14244236

**Published:** 2025-12-10

**Authors:** Andrea Marianelli, Cecilia Akotowaa Offei, Monica Macaluso, Nicola Mercanti, Bruno Casu, Angela Zinnai

**Affiliations:** 1Department of Agriculture, Food and Environment, University of Pisa, Via del Borghetto 80, 56124 Pisa, Italy; andrea.marianelli@phd.unipi.it (A.M.); ceciliaakotowaa.offei@unito.it (C.A.O.); nicola.mercanti@phd.unipi.it (N.M.); angela.zinnai@unipi.it (A.Z.); 2Department of Agricultural, Forest and Food Sciences, University of Turin, Via Verdi 8, 10124 Torino, Italy; 3INFN Pisa Section, Largo Bruno Pontecorvo 3, 56127 Pisa, Italy; bruno.casu@cern.ch; 4European Laboratory for Nuclear Research (CERN), 1211 Geneva, Switzerland; 5Interdepartmental Research Center “Nutraceuticals and Food for Health”, University of Pisa, 10 Via del Borghetto 80, 56124 Pisa, Italy

**Keywords:** computational, fermentation, stability, prediction, microstructure, sustainability, optimization, data driven

## Abstract

Traditional shelf-life (SL) determination in bakery products relies primarily on subjective sensory evaluation, limiting both predictive capability and technological transfer. This study aimed to develop an objective, data-driven framework by integrating statistical and Machine Learning (ML) methods to identify and quantify the core determinants of bread SL. Samples were produced under a 2 × 2 × 2 factorial design (Fermentation, Temperature, Packaging), with continuous monitoring of physicochemical and atmospheric parameters. Three-way ANOVA confirmed that Storage x Temperature (η^2^ ÷ 0.41) and Modified Atmosphere Packaging (η^2^ ÷ 0.36) were the dominant factors. The optimal synergy (4 °C + ATM) achieved a 100% Success Rate, extending SL to 54 days vs. 16 days under ambient conditions. For prediction, a Generalized Linear Model (GLM) was developed for binary classification and rigorously validated via 10-fold cross-validation. The GLM achieved an Overall Accuracy of 89% (AUC 92%), uniquely identifying pH and Total Titratable Acidity (TTA) as the most influential predictors. In conclusion, GLM provides a robust tool for objective SL prediction. The integrated ANOVA–GLM framework achieved a 3.3-fold SL extension and 92% predictive accuracy. The findings confirm that preservative effectiveness is not solely due to the process itself, but is mediated by the resulting chemical acidity, offering a scalable framework for Real-Time Quality Control (QC) in the food industry.

## 1. Introduction

Sourdough is increasingly interpreted not merely as a traditional leavening system, but as a biologically active preservation mechanism that can conditionally modulate the deterioration trajectory of bakery products. While conventional yeast fermentation primarily contributes to gas retention and flavor development, PDO sourdough generates a broader metabolite profile capable of restructuring moisture migration, acidification dynamics, and microbial ecology at the matrix level [1,2,3]. This shift in perspective reframes sourdough from a formulation choice to a context-dependent technological system, whose stabilizing effect is activated only under specific environmental constraints, rather than exerted universally.

Classical food preservation theory provides the conceptual foundation for this conditionality. According to hurdle technology, stability does not emerge from any single barrier in isolation but from the synergistic activation of multiple interdependent constraints [4]. Within this framework, PDO systems behave as metabolic hurdles, whose bio-preservative function is expressed only when the surrounding regime enables their biochemical advantage over spoilage dynamics [5,6]. This implies that sourdough cannot be evaluated as a static “ingredient effect”, but as a threshold-governed process whose efficacy depends on whether storage conditions allow its metabolic dominance to manifest.

Recent literature reinforces this view by demonstrating that shelf-life extension in sourdough-based matrices derives not from microbial suppression alone, but from a shift in the dominant failure mode from fungal spoilage to structural degradation [7,8,9]. In other words, PDO does not merely delay mold onset: it reorganizes the deterioration trajectory so that the limiting factor of product acceptability transitions from microbiological to textural collapse. This trajectory-based interpretation has rarely been captured quantitatively, because most previous studies have compared endpoints descriptively rather than inferring the mechanism that governs the SL outcome.

A mechanistic evaluation, rather than a descriptive one, requires identifying whether PDO has a direct effect on SL, or its effect is entirely mediated by physicochemical shifts on acidification (pH, TTA) and organic acid ratios. Classical statistical approaches only partly address this question: ANOVA can detect significant differences between conditions but does not determine whether a factor is causal or mediated. ML approaches, when used interpretively rather than as black-box prediction tools, provide a methodological bridge for linking environmental conditions, metabolic activation, and shelf-life outcome [10,11,12]. GLMs provide a formal statistical bridge between categorical factorial design and probabilistic inference. Their logit formulation allows identification of latent mediators (e.g., pH, TTA), aligning with frameworks proposed by [13].

In this context, GLM-based inferential models are particularly suited to small experimental systems where preservation is governed by threshold effects and conditional stability [13,14,15]. Unlike tree-based classifiers, GLMs allow mediation to be explicitly tested, enabling acidification to be evaluated not as a correlated descriptor, but as the latent mechanism through which PDO achieves SL extension [16,17,18]. When combined with PCA to assess variance structure, this framework allows preservation to be interpreted as a system-level transition rather than a univariate improvement. Accordingly, this study investigates whether PDO sourdough functions as a threshold-dependent bio-preservation system, identifies the storage regime under which this activation emerges, and quantifies the mediating physicochemical pathway that governs the shift in deterioration trajectory.

## 2. Materials and Methods

### 2.1. Experimental Design

A full-factorial design (2 fermentative systems × 2 storage temperatures × 2 packaging conditions, *n* = 4 replicates per condition; total *n* = 32) was adopted to identify the environmental threshold at which PDO activates a technological bio-preservation function. The two spoilage-governing fermentative systems were: baker’s yeast fermentation (Control), and sourdough fermentation (PDO). Storage conditions were imposed at 4 °C or 20 °C, under air or atmospheric packaging (ATM; 100% CO_2_). The PDO sourdough used in this study was prepared according to the official Tuscan Bread PDO protocol, following the multi-stage refreshment and fermentation protocol detailed in the following section.

### 2.2. Breadmaking Process and Sampling Schedule with Analytical Frequency

The samples (*n* = 4 for each condition) were prepared following a standardized dough formulation expressed on a flour basis (baker’s percentage): Bologna Type 0 flour 100% (milled by Molino Angeli Srl, Pietrasanta, Italy), distilled water 75%, sourdough 14% (Tuscan Bread PDO Consortium, Arezzo, Italy). Control doughs were leavened with baker’s yeast, whereas PDO samples were produced using the traditional Tuscan Bread PDO sourdough, both sharing an identical base formulation.

The sourdough starter was refreshed according to the protocol established by the Tuscan Bread PDO Consortium [19] and in particular, the starter preparation (biga) provides the mixture of its central portion with an equal amount of strong wheat flour (W > 260) and 6 g of fructose, and the consequent addition of demineralized water to achieve 42–48% hydration relative to the flour weight. All the ingredients were mixed for 15 min until the consistency of the dough was obtained. Then, the dough was stretched and folded 5–6 times, shaped into a sphere, cross-cut on the surface, and placed in a sealed container. Finally, the dough was left to ferment in a thermostatic chamber set at 28 °C for 4 h. This process was repeated twice. After the second refreshment phase, the biga dough was prepared. An aliquot of 100 g of twice refreshed sourdough starter was mixed with strong wheat flour (W > 260). Then, water was added in an amount equal to 60% of the flour weight. Afterwards, the mixture was mixed for 15 min until the dough was consistent, and finally, the dough was placed in a glass bowl covered with food-grade plastic wrap in a thermostatic chamber set at 17 °C for 16 h.

Before mixing the dough, a short autolysis was performed to activate endogenous wheat enzymes and facilitate starch hydrolysis into fermentable sugars. Two-thirds of the total water was combined with the flour and mixed for 15 min with a kneading machine (Model Grilletta IM 5S/230 10 V HH, Famag Srl, Molise, Italy), followed by 30 min of resting at room temperature. The remaining water and 225 g of pre-fermented dough were then incorporated, and all ingredients were mixed until a smooth and homogeneous structure was obtained.

The final doughs were fermented in a thermostatic chamber at 38 °C for 90 min (model FOALSTR23M, Fimar S.p.A., Villa Verrucchio, Italy), subsequently divided into 460 g loaves, and proofed for 120 min at 38 °C and 86 RH% on wooden boards covered with linen cloths. Baking was performed at 180 °C for 50 min in a ventilated oven (model FOSTR1040T, Fimar S.p.A., Villa Verrucchio, Italy). The internal temperature of the bread loaves was monitored until 96 ± 2 °C was reached using a digital penetration thermometer (Model Checktemp HI98501, Hanna Instruments Inc., Woonsocket, RI, USA) immediately after baking and during the cooling phase to ensure consistent thermal profiles across replicates. Baked samples were cooled for 2 h under a laminar flow hood (model Olympia 1.2, Bioair instruments S.r.l., Siziano, Italy) until 22 ± 2 °C internal temperature was reached (Figure 1). All kneading operations were carried out using a kneading machine (Model Grilletta IM 5S/230 10 V HH, Famag Srl, Molise, Italy). Mixing lasted approximately 10 min until the dough reached a smooth and elastic consistency.

After the breadmaking process, the samples were packed into PET (polyethylene) bags filled with either technical air or a modified atmosphere (MAP) composed of 100% *v/v* CO_2_. The samples were stored. Moreover, to avoid artifacts associated with repeated opening of the same unit, all samples were packaged individually, stored at room temperature (20–24 °C) and 40 RH%, and analyzed as destructive replicates. Each measurement, therefore, represents an independent unit withdrawn at a specific time point, ensuring that packaging integrity, headspace conditions, and moisture dynamics were not perturbed by sampling.

The analytical frequency was adapted to the physiological trajectory of deterioration. During the early phase (0–16 days), spoilage risk was low, so sampling was performed every 4 days. Between 16 and 30 days, the divergence between treatments emerged; thus, analyses were conducted every 6–8 days. Beyond day 30, monitoring continued every 10–12 days until the end of SL (54 days).

However, their terminal failure mode remained fungal-driven, unlike PDO samples that transitioned to a texture-limited endpoint (Table 1).

This adaptive schedule ensured high temporal resolution around the mechanistic threshold while preserving packaging integrity and avoiding pseudo-replication. Indeed, to maximize the temporal resolution of GLM mediation analysis, sampling intervals were selected to increase data density in proximity to the expected deterioration threshold, while minimizing redundant early-stage measurements. A destructive sampling strategy was adopted because repeated non-destructive evaluations on the same loaf introduced variability due to mechanical perturbation, moisture redistribution, cut-surface exposure, and structural fatigue. Independent destructive units, therefore, ensure higher measurement precision and eliminate within-unit correlation effects that would otherwise bias GLM inference. It also reflects a biologically realistic sampling strategy, aligned with the intrinsic time of spoilage acceleration once metabolic or environmental constraints are exceeded.

### 2.3. Physicochemical Monitoring

For each fermentation × temperature × atmosphere combination, four independent loaves (*n* = 4) were produced and treated as separate experimental units. During baking, the position of loaves within the oven was rotated across batches to minimize positional effects. After cooling, the loaves were randomly assigned to storage containers and sampling time points. The order of physicochemical measurements (pH, Aw, TTA, organic acids, texture) was also randomized across samples to avoid systematic analytical bias. Variability between units is reported as standard deviations (±SD) and was incorporated directly at the replicate level in ANOVA and GLM analyses. The loaves were analysed at each scheduled time point (as described in Table 1 in the previous section). All instruments were calibrated prior to each analytical session: pH electrodes using standard buffers at pH 4.00 and 7.00, the Aw-meter using manufacturer-supplied reference salts with automatic temperature compensation, and TTA titration following AOAC-standardized NaOH calibration. Duplicate measurements were performed when relevant to ensure consistency and reduce analytical error.

#### 2.3.1. pH Physicochemical Determination

The pH value was measured on 10 g of crumb dispersed in 90 mL of distilled water, using a calibrated glass electrode, according to AACC Method 02–52.

#### 2.3.2. Aw Physicochemical Determination

Aw was determined using a HygroPalm HP23-AW-A instrument (Rotronic AG, Bassersdorf, Switzerland) at 25 ± 0.2 °C, after equilibrium stabilization of the crumb surface, ensuring that measurements reflected the internal moisture gradient.

#### 2.3.3. TTA Physicochemical Determination

TTA was determined as described by [20]. Ten grams of the sample were homogenized in 90 mL of distilled water and titrated with 0.1 N NaOH to pH 6.5. Results were expressed as mL NaOH 0.1 N per 10 g of sample.

#### 2.3.4. Fermentative Metabolites Determination

Lactic and acetic acids were quantified enzymatically using Megazyme kits (Megazyme Ltd., Wicklow, Ireland) following the manufacturer’s instructions. Results were expressed as g/100 g dry matter.

#### 2.3.5. Texture Analysis

The compressibility of bread slices was evaluated using a PNR-12 penetrometer (Anton Paar, Rivoli, Italy). Each loaf was analysed destructively: two central slices per sample were tested, and three penetration points were measured per slice using a 40 g load applied for 5 s. The instrument records the penetration force as Newtons (N), which directly represents the resistance of the crumb to compression. The average of six readings per sample (2 slices × 3 points) was used for statistical analysis. The textural-collapse threshold was not chosen arbitrarily. The end of the shelf life and textural collapse was defined by the end of bread shelf life by a textural analyzer, such as increasing hardness, decreasing springiness, and cohesiveness, indicating staling or textural collapse. These objective measurements were based on preliminary internal tests and literature on bread texture limits for shelf-life assessment and aligned with consumer rejection and loss of bread quality [21,22,23,24].

All physicochemical parameters were measured in quadruplicate (*n* = 4) per condition and expressed as mean ± standard deviation. These values served as continuous predictors in the GLM model framework (Section 2.4 below).

These measurements served as mechanistic intermediates linking PDO metabolism to spoilage trajectory.

### 2.4. ML Model (GLM Approach)

A GLM, binomial family with logit link, Equation (1) [15], was implemented to test whether PDO acts as a direct categorical factor or through physicochemical mediation. The model was coded and built in R (R Core Team, Version 4.3.0, Vienna, Austria) using the *stats* and *caret* packages. To evaluate whether shelf-life acceptability was directly determined by the fermentation type or indirectly mediated through physicochemical shifts, a GLM was implemented.

The model expressed the probability (*p*) that a sample remained acceptable (no visible fungal growth and texture > 3.3 N) as a function of key physicochemical predictors:

The dataset consisted of 32 observations (8 storage × 4 fermentation conditions, with 4 replicates each), with shelf life encoded as a binary outcome (acceptable/non-acceptable) (Figure 2). Therefore, samples were classified as texturally acceptable when crumb firmness remained below 3.3 N and no visible spoilage was detected. This threshold reflects the limit at which structural collapse becomes perceptible both instrumentally and sensorially, marking the transition from an elastic crumb to a compact, starch-retrograded matrix. Maintaining firmness below this value is consistent with previously reported consumer rejection limits for bread and ensures that textural degradation is detected before microbiological spoilage becomes dominant. The combined use of mechanical and visual criteria provides a robust, conservative definition of acceptability that supports the subsequent GLM-based classification framework. The choice of four replicates per condition was based on ensuring sufficient sensitivity to detect interaction effects and threshold-dependent activation, while maintaining full independence of observations under destructive sampling. Power considerations focused on medium to large interaction magnitudes typical of fermentation to environment systems, making *n* = 4 the optimal balance between inferential resolution and feasibility. The destructive sampling approach minimized intra-sample variability by avoiding repeated mechanical or physicochemical perturbations of the same loaf, which are known to introduce noise in texture, Aw, and pH measurements. Each value, therefore, reflects an unaltered physical state, improving the accuracy of GLM coefficient estimation.
(1)$$\mathrm{logit}(p)=ln(\frac{p}{1-p})=\beta_{0}+\beta_{1}(pH)+\beta_{2}(TTA)+\beta_{3}(Lactic)+\beta_{4}(Acetic)$$ where:
*β*_0_ is the intercept;*β*_1_*–β*_4_ are the estimated regression coefficients describing the contribution of each variable to the log-odds of shelf-life acceptability.

**Figure 2 foods-14-04236-f002:**
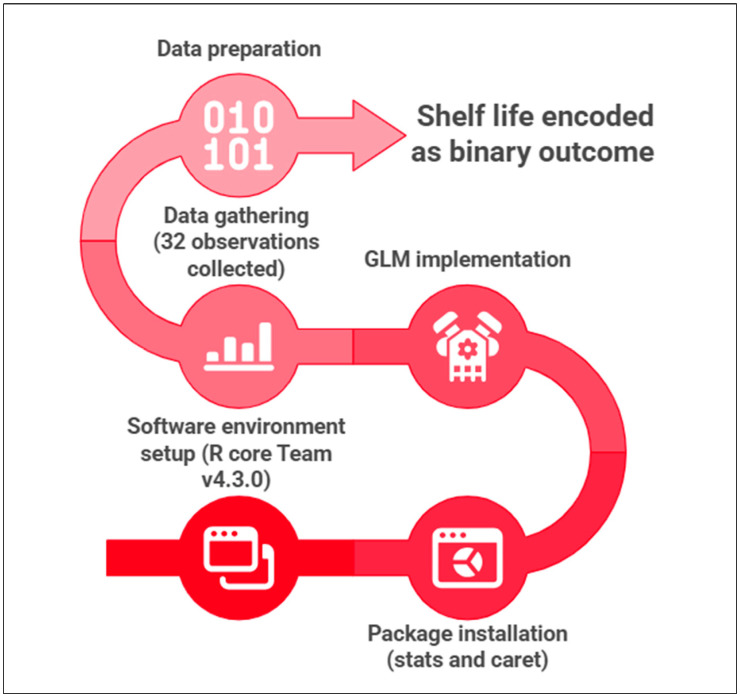
Workflow of the GLM-based modeling pipeline used to mechanistically validate PDO SL behavior.

To stabilize inference and evaluate mediation, the GLM was reformulated using physicochemical parameters (pH, TTA, lactic acid, acetic acid) as predictors. Variable selection was performed via stepwise AIC [25], and model robustness was assessed using 10-fold cross-validation [26]. This procedure allowed the GLM to act as an inferential bridge from factor dominance (ANOVA) to latent mediation (PCA), confirming that PDO becomes predictive only once its acidification profile is expressed. The model performance was reported in terms of overall accuracy, recall, and AUC (see Table 2 in Section 3.5).

### 2.5. Statistical Analysis

Data was analyzed using a two-stage inferential pipeline combining traditional and data-driven modeling. A full-factorial three-way ANOVA (F × T × A) was shown in Equation (2) and applied to identify dominant preservation factors and their interactions. Parameters like pH, TTA, lactic and acetic acids, lactic/acetic ratio, water activity, and texture were tested for normality (Shapiro–Wilk) and homoscedasticity (Levene–Brown–Forsythe).
(2)$$\mathrm{three}-\mathrm{way}\;\mathrm{ANOVA}\;\mathrm{model}:\;Y_{ijk}=\mu +F_{i}+T_{j}+A_{k}+(F\times T{)}_{ij}+(F\times A{)}_{ik}+(T\times A{)}_{jk}+\varepsilon_{ijk}$$ where:
$Y_{ijk}$ represents the observed value of the dependent variable under the *i*-th fermentation type, *j*-th temperature, and *k*-th atmosphere condition;μ is the overall mean;$F_{i}$,
$T_{j}$, and
$A_{k}$ are the main effects of Fermentation, Temperature, and Atmosphere;$(F\times T{)}_{ij}$,
$(F\times A{)}_{ik}$, and
$(T\times A{)}_{jk}$ are the two-way interactions among these factors;$\varepsilon_{ijk}$ is the random error term, assumed to be normally distributed with zero mean and homogeneous variance (
$\varepsilon_{ijk}∼N(0,\sigma^{2})$).

Tukey’s HSD post hoc test (α = 0.05) was used for pairwise contrasts, and effect sizes were expressed as partial η^2^ (Equation (3)).

Equation (3): Effect sizes were computed as partial eta squared ($\eta^{2}$) according to [27]
(3)$$\eta^{2}=\frac{SS_{\mathrm{factor}}}{SS_{\mathrm{total}}}$$ providing a measure of the relative contribution of each factor to total variance.

This factorial structure allows the identification not only of single-factor influences (main effects) but also of synergistic or antagonistic interactions among environmental (T × A) and biological (F × T) parameters that condition the activation of PDO’s preservative effect.

To visualize multivariate covariance and relationships between chemical variables and SL regimes, a PCA was performed for structure exploration (JMP^®^ 19, Cary, NC, USA: SAS Institute Inc.). Each replicate (*n* = 4 per condition) was included as an independent observation in the PCA rather than averaging across loaves, to retain within-condition variability and avoid artificially inflating between-group separation. This approach preserves the natural dispersion structure of physicochemical data. Subsequently, a GLM was implemented to mechanistically validate whether PDO acts as a direct categorical factor or through physicochemical mediation. The model was coded in R (R Core Team, Version 4.3.0, Vienna, Austria) using the *stats*, *caret*, and *pROC* packages. The dataset comprised 32 observations (8 combinations × 4 replicates), with SL encoded as a binary outcome (1 = acceptable, 0 = non-acceptable). An initial model using only categorical predictors (F × T×A) produced complete separation; the PDO + 4 °C/Air condition yielded perfect success [15,28], confirming a threshold-dependent activation mechanism. A survival analysis approach (e.g., Kaplan–Meier or Cox modelling) was not applicable in this study because shelf-life assessment was destructive. Each loaf was measured only once and discarded, preventing the reconstruction of individual deterioration trajectories across time. Therefore, time-to-failure curves and hazard functions could not be estimated. For this reason, shelf-life was represented as a binary state (acceptable vs. collapsed) at each sampling point, which is the appropriate approach for destructive experimental designs. To overcome this, the GLM was refitted using physicochemical mediators (pH, TTA, lactic, and acetic acid) as predictors. Variable selection employed stepwise AIC optimization and 10-fold stratified cross-validation [26].

## 3. Results and Discussion

### 3.1. Redefinition of the Spoilage Trajectory

SL results showed a clear divergence between the PDO and the yeast-fermented samples, demonstrating that preservation emerged not as a general improvement but as a change in the limiting mechanism of deterioration. The control samples (yeast fermentation) reached the end of SL at day 16 under both refrigeration and ambient storage, whereas PDO samples remained acceptable up to day 54 in the refrigerated air conditioning. The endpoint was defined by a dual criterion: firstly, visible fungal growth and, secondly, irreversible textural collapse measured via the PNR12 penetrometer, which aligns with the failure-mode definition of acceptability in bakery matrices.

Importantly, PDO did not merely delay fungal colonization, but replaced fungal spoilage as the limiting failure mode with structural degradation as the downstream constraint. This observation is consistent with recent evidence indicating that sourdough-based systems alter the deterioration trajectory, shifting the product from a microbially terminated SL toward a structure-governed one [29]. Under this interpretation, the extension is not the effect itself, but the consequence of a transition in the dominant deterioration pathway.

The magnitude of the difference (16 to 54 days) cannot be explained by microbial growth delay alone. Instead, it suggests the presence of a threshold-based activation mechanism in which PDO becomes metabolically dominant over spoilage flora only when the storage regime supports its biochemical advantage [30,31]. This finding anticipates the mechanistic analysis developed in the subsequent sections, where physicochemical data and multivariate modeling confirm that the extended shelf-life is mediated by acid-driven restructuring of the matrix rather than by antifungal inhibition alone.

### 3.2. Physicochemical Mediation Emerges Before SL Outcome

PDO fermentation produced a markedly different biochemical environment compared to yeast fermentation, preceding and mechanistically underpinning the observed divergence in SL. PDO samples showed significantly lower pH values and markedly higher TTA, together with an increased accumulation of lactic and acetic acids, confirming that acidification is not a coincidental correlation but the metabolic driver enabling the shift in deterioration trajectory. These changes appeared before visible spoilage divergence, indicating that PDO modifies the stability regime at a biochemical rather than microbiological endpoint.

The modulation of the Lactic acid/Acetic acid ratio further supports this interpretation: PDO profiles were dominated by a more efficient conversion towards organic acids, restructuring the matrix into a metabolically more stable state. This aligns with recent evidence that sourdough preservation effectiveness is governed by metabolic activation thresholds and not by the mere presence of LAB cells [32,33,34]. In this framework, PDO does not act as an antimicrobial barrier, but as a biochemical condition-setter, shifting the equilibrium of the system toward a lower pH/high acid environment that constrains the ecological feasibility of fungal proliferation.

These physicochemical trends anticipate the outcome-level differences reported in Section 3.1 by demonstrating that metabolic mediation precedes visible divergence in deterioration. By presenting a stable and highly acidified profile, PDO establishes the precondition for SL extension, which is later expressed as a transformation in the dominant failure mode [35,36,37]. Table 2 summarizes these physicochemical parameters, while Figure 3 graphically highlights the differences and their accumulation relative to controls.

### 3.3. Environmental Dominance in ANOVA Supports a Threshold-Mediated Mechanism

The three-way ANOVA confirmed that environmental factors such as Temperature (T) and Atmosphere (A) significantly affect SL than the fermentative system (F) itself (Table 3).

T and A displayed the highest effect sizes (η^2^), while the main effect of fermentation on SL was not statistically significant. This pattern indicates that PDO does not act as a direct preservation factor, but exerts its stability advantage only when the environmental regime activates its biochemical dominance. In other words, PDO is conditionally effective, not universally active.

The significant F × T interaction further reinforces the interpretation that PDO requires a specific thermal regime to express its preservation capacity. This observation is consistent with SL theory, where the limiting mechanism of stability is governed by the dominant external constraint until a transition threshold is crossed [38,39]. Recent work has demonstrated that, in matrix-governed deterioration systems, formulation alone rarely determines stability unless it interacts with temperature-driven barriers [40,41]. Within this logic, environmental parameters set the feasibility domain of PDO activity, while fermentation defines the metabolic trajectory inside that domain.

Accordingly, the ANOVA results anticipate mediation: if temperature and atmosphere are the dominant explanatory factors, but the final SL behavior is only observed in PDO systems. Table 3 shows that the preservation advantage of PDO is not simply due to being a sourdough-based fermentative system, but emerges when the storage regime supports its metabolic activity. This set the mechanistic premise for the PCA presented in Section 3.4, which quantified whether physicochemical variables represented the latent structure through which PDO expressed its preservation effect.

### 3.4. PCA Reveals Acidification as the Latent Structure Governing SL

Prior to PCA, all variables were mean-centered and scaled to unit variance (autoscaling) to prevent variables with larger magnitudes from dominating the solution. Each replicate (*n* = 4 per condition) was included as an individual observation in the analysis. The PCA confirmed that physicochemical variation in the dataset is not distributed randomly but is organized along a single dominant latent axis corresponding to acidification and organic acid accumulation (Figure 4). Furthermore, potential batch effects were evaluated by color-coding PCA score plots according to analytical session, measurement day, and instrument batch. No clustering attributable to session-related artefacts was observed, indicating that the multivariate structure reflects true biological and process-driven variability rather than experimental drift.

PC1 alone accounted for 97.8% of the total variance, with positive loadings for TTA, lactic, and acetic acids, and a strong negative loading for pH. This structure indicates that the differentiating mechanism between PDO and yeast systems is not categorical, but metabolic; the intensity of acidification defines the stability domain.

The clear separation between PDO and yeast samples along PC1 supports the hypothesis introduced in Section 3.3: PDO does not act as a universal or formulation-driven effect, but as an activation-dependent metabolic state. In other words, fermentation type is not the explanatory variable, but the biochemical shift generated by PDO is. This latent clustering is consistent with the mechanistic interpretation of sourdough preservation emerging in literature, where stability results from a metabolic restructuring of the matrix rather than additive antifungal inhibition [1,7,42,43]. Importantly, PCA provides the missing causal bridge between ANOVA and GLM.

ANOVA showed that environment dominates (threshold-setting), PCA shows that acidification is the latent mediator, hence GLM must evaluate preservation through physicochemical variables, not categorical fermentation.

This is also why complete separation later emerges in the binary classification model (Section 3.5). Once samples cross the metabolic activation threshold defined by PC1, they fall into a different deterioration domain altogether. PCA, therefore, does not merely visualize clustering but confirms that PDO’s preservation capacity is mediated and is structurally encoded in the acidification axis that governs variance in the system. To evaluate potential batch effects, score plots were inspected according to analytical session, and no systematic clustering by batch was observed. Therefore, no additional batch correction was applied.

### 3.5. GLM Modeling Confirms Mediation and Explains Complete Separation

The GLM model was developed to test whether the SL outcome was explained by fermentation type directly or by the physicochemical state generated by PDO metabolism to lay the know-how basis for an ML-nondestructive system. Although cross-validation supports internal consistency, external validation on larger datasets is necessary to assess the generalizability of the GLM decision boundaries. Such validation would allow testing the stability of the estimated coefficients against broader sources of biological and technological variability, including batch-to-batch differences in sourdough performance. When the model was initially fitted using only categorical experimental factors (F × T × A), it failed due to complete separation because PDO samples stored at 4 °C in ATM formed a perfect success class. This phenomenon is not indicative of overfitting, but a known property of logistic models in which the predictor structure deterministically separates the outcome domain [15,28]. In this case, separation emerged because the effect of PDO is not probabilistic but latent and mediated: once acidification crosses the activation threshold, samples transition into a different deterioration regime altogether. Although the GLM showed strong internal coherence, the limited dataset (*n* = 32) increases the uncertainty surrounding the stability of the estimated decision boundaries. The emergence of complete separation reflects the deterministic structure of the physicochemical variables rather than overfitting, but validation on larger, independent datasets is required to quantify predictive variability.

To stabilize the model and move from categorical to mechanistic inference, the GLM was reformulated using physicochemical variables (pH, TTA, lactic, and acetic acid) as predictors [44,45,46]. After AIC-based selection [25], the final model was validated via 10-fold cross-validation [26], achieving an overall accuracy of 0.89, a recall of 0.915, and an AUC of 0.92 (Table 4). These results suggest that preservation was not an intrinsic property of PDO as a factor, but a consequence of the biochemical state it induces. This supports the mediation pathway established by the PCA, where PDO drives the latent shift in acidification, and it is this metabolic that predicts stability.

Accordingly, the GLM confirms the mechanistic role of PDO as a threshold-dependent activator of preservation [47]. Fermentation type becomes predictive only once its metabolic state is expressed, meaning that SL extension is statistically downstream of physicochemical mediation [36]. This result closes the inferential loop initiated in Section 3.3, where ANOVA identifies environmental dominance, PCA reveals the latent axis, and GLM validates that the PDO effect is dynamically activated, not statically present [48].

### 3.6. From Chemical Factors to the Mediation and Activation

Taken together, the results may indicate that sourdough (PDO) does not extend SL as a direct categorical factor, but as a threshold-activated metabolic system. The ANOVA established that environmental constraints (T × A) are the dominant determinants of stability, while fermentation alone is insufficient to explain the preservation outcome. The PCA then confirmed that PDO operates through a latent axis of acidification once physicochemical restructuring is established; samples cluster into a metabolically stable domain. Finally, the GLM validated that SL is predicted not by fermentation type per se, but by the metabolic state it induces, as evidenced by the emergence of complete separation when categorical variables were used.

Thus, PDO functions as a conditional preservation mechanism whose efficacy is activated only under a supportive storage regime, meaning that stability is mediated rather than inherent. Preservation arises when the environmental barrier enables PDO to express its metabolic dominance, linking the statistical evidence to a mechanistic interpretation: environment sets the feasibility domain, PDO defines the trajectory within it. Lastly, to synthesize the inferential findings into a practical and industrially interpretable framework, a decision tree was developed to summarize the outcome of the ANOVA–GLM sequence. Following is shown in Figure 5 the model built in R (R Core Team, Version 4.3.0, Vienna, Austria) using the *rpart* and *rpart.plot* packages on the full dataset (*n* = 32). The hierarchical structure visualizes threshold-dependent rules: lactic acid, TTA, and pH are the key variables governing the classification, while texture contributes as a secondary discriminant. Terminal nodes report group identity (PDO or Control), number of samples (*n*), and classification accuracy (%). This decision tree shows the outcome of the GLM, translating the statistical mediation model into interpretable decision rules. Based on the model, PDO activation follows a conditional metabolic rule rather than a categorical effect. The preservation advantage emerges only when the biochemical state of the dough crosses specific physicochemical thresholds of acidification and buffering capacity.

The following quantitative rule summarizes this activation mechanism (Equation (4)):
(4)$$\left(\mathrm{Lactic}\geq 0.18\;g/100\;g\right)\;\wedge \left(\mathrm{TTA}\geq 0.03\;\mathrm{meq}/g\right)\;\wedge \;\left(\mathrm{pH}<4.2\right)=\mathrm{PDO}-\mathrm{active}\;\mathrm{stability}\;\mathrm{domain}$$


These rules indicate that PDO fermentation becomes functionally active as a bio-preservation system only when lactic acid accumulation and total titratable acidity exceed critical thresholds while pH drops below the inhibitory limit for fungal proliferation. The numerical thresholds reported in Equation (4) were directly generated by the decision tree during model training. These split values represent the points where the model best separated breads that maintained acceptable texture from those that deteriorated. We verified the robustness of these thresholds by repeating the tree fitting procedure under different random seeds, observing only minor numerical fluctuations and no changes in the overall decision structure. This confirms that the thresholds reflect stable patterns in the data rather than random variability.

Practically, this provides operational decision criteria for QC. Once these biochemical parameters are reached, the product transitions from a microbially limited to a structure-limited deterioration regime.

This threshold-based framework supports the design of real-time monitoring protocols and can be implemented in machine-learning-based SL prediction systems.

Indeed, confirming the threshold-dependent activation of PDO metabolism inferred from the GLM. PDO fermentation dominates the preservation regime, extending SL through matrix restructuring rather than direct anti-fungal inhibition.

## 4. Conclusions

Previous work has frequently reported SL improvements in sourdough systems, but nearly all studies have treated sourdough as a static formulation variable rather than as a conditionally activated mechanism. Because these studies did not distinguish presence from function, nor compare preservation across different ecological regimes, the threshold behavior governing PDO activation remained concealed. As a result, sourdough was interpreted as universally preservative when it is, in fact, mechanistically contingent.

This study suggests that PDO fermentation operates as a conditionally activated preservation system, whose activity depends on environmental feasibility. Through ANOVA, we showed that storage conditions (T × A) dominate the stability outcome, confirming that PDO is not directly effective as a categorical factor. The PCA established that preservation emerges through a latent physicochemical axis defined by acidification (pH/TTA), and the GLM validated that SL is predicted only after mediation is accounted for, explaining the emergence of complete separation as a structural rather than probabilistic effect. Functionally, this metabolic activation translated into a 3.4-fold extension (16 to 54 days), not as a slower version of the same deterioration process, but as a change in the dominant failure mode.

Technologically, these findings reposition PDO as a process variable to be managed as a controllable metabolic state, rather than as a formulation ingredient. PDO appears to become operational only when refrigeration constrains the ecological feasibility of fungal growth, suggesting that SL control may need to be expressed through acidification targets rather than recipe inclusion alone. The small-N design is not a limitation but the appropriate epistemic framework for mediation testing.

This design is optimal for causal mediation testing but limits the generalization of the GLM model across different product types or industrial batches. The limited dataset also increases uncertainty in model-based estimates, and independent validation across different production cycles is required to confirm the stability of the predictive patterns observed herein.

Therefore, the present findings should be considered promising but preliminary. Future work should validate these patterns across larger industrial batches, allowing the extension of this inferential framework to large-N datasets and enabling the use of ensemble learning models (e.g., Random Forest, Support Vector Machines) combined with nested cross-validation to verify the stability and generalizability of the predictive structure.

Additionally, while the present work indicated PDO activation as a threshold-dependent metabolic mechanism, further investigations integrating metabolomic and microbial genomic profiling could elucidate the biochemical pathways underlying the acidification-mediated stabilization process. These observations align with recent evidence showing structural stabilization driven by biochemical modulation in food matrices, including polysaccharide-mediated barrier enhancement [49] and protein structural adaptations under processing stress [50], reinforcing the mechanistic interpretation proposed herein.

This would strengthen the transition from empirical bio-preservation toward predictive, data-driven quality control systems suitable for industrial-scale deployment.

## Figures and Tables

**Figure 1 foods-14-04236-f001:**
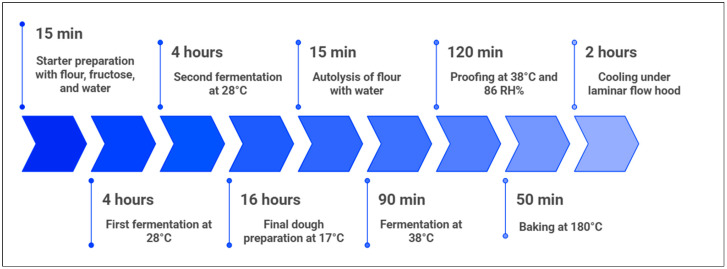
Breadmaking process flow chart: from refreshment to the cooled samples.

**Figure 3 foods-14-04236-f003:**
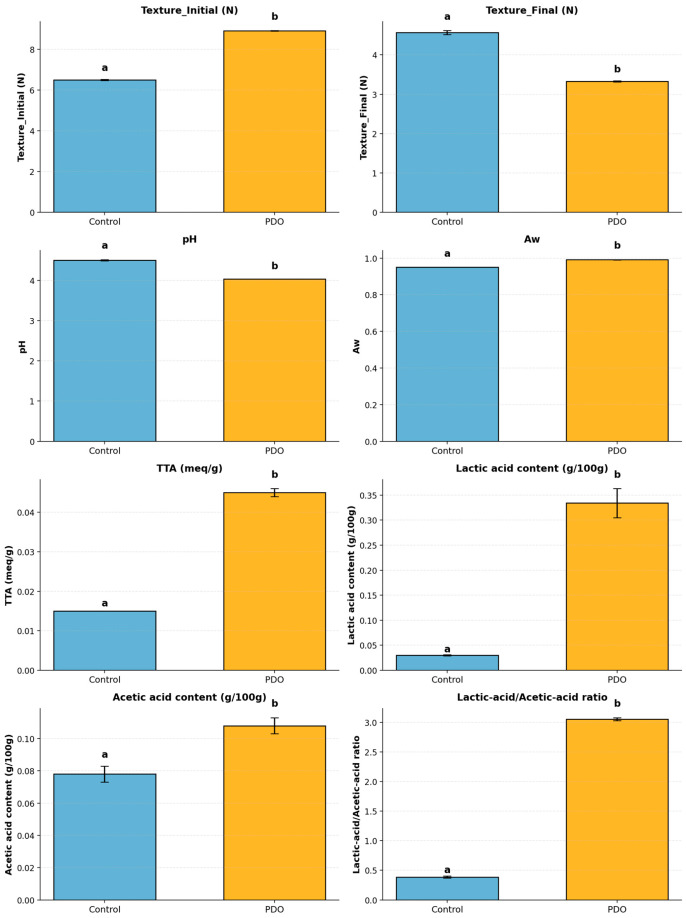
Bars represent standard deviation (*n* = 4) for physicochemical and textural properties of Control (Yeast) and PDO (Sourdough) bread samples. Different superscript letters (a,b) in each row indicate significant differences between treatments according to Tukey’s HSD post hoc test (*p* < 0.05).

**Figure 4 foods-14-04236-f004:**
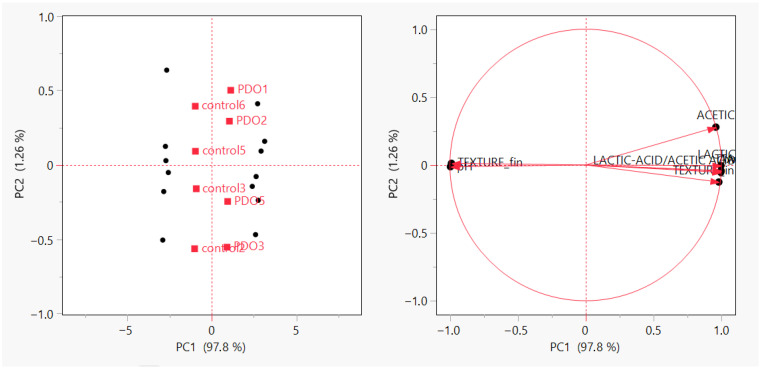
Principal Component Analysis (PCA) of physicochemical parameters in Control (Baker’s yeast) and PDO (Sourdough) bread samples ((**Left**): PCA score plot showing sample distribution; (**Right**): loading plot illustrating variable contributions).

**Figure 5 foods-14-04236-f005:**
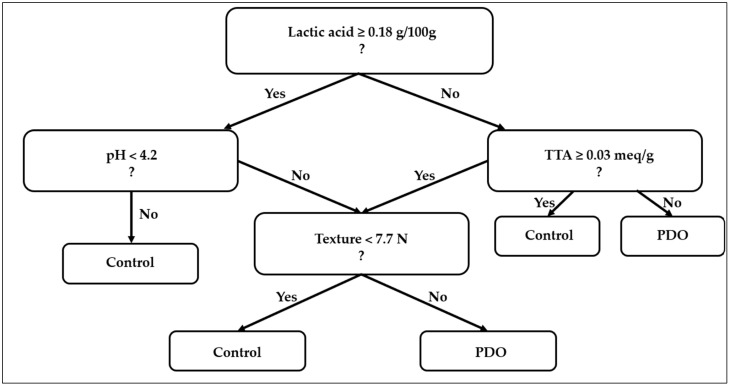
Decision tree summarizing physicochemical predictors of SL classification between PDO (sourdough) and Control (yeast) bread samples. *Root*
*node:* starting variable defining global separation (lactic acid). *Internal nodes:* secondary discriminants based on pH, TTA, or texture thresholds. *Terminal nodes:* predicted class (PDO or Control) with sample count (*n*) and classification accuracy (%). *Yes/No branches:* logical decisions defining the threshold direction for each split. ?: decisional root tree from Random Forest Decisional Tree.

**Table 1 foods-14-04236-t001:** Sampling schedule and analytical frequency for control and PDO samples across the full storage period (0–54 days). Each data point corresponds to an independent destructive unit.

Storage Phase (Days)	Frequency (Days)	Units Analysed	Quality Status at Sampling
Early storage phase (0–16)	Every 4	Control + PDO	Stable
Mid-storage phase (16–30)	Every 6–8	Control + PDO	Initial Fungal Growth
Late storage phase (30–54)	Every 10–12	PDO + Control ATM	Hardening texture-limited, Fungal Growth

**Table 2 foods-14-04236-t002:** Physicochemical and textural characteristics of Control (Baker’s yeast) and PDO (Sourdough) bread samples (mean ± SD, *n* = 4).

Parameter	Control (Mean ± SD)	PDO (Mean ± SD)
Texture_Initial (N)	6.494 ± 0.030 ^a^	8.910 ± 0.010 ^b^
Texture_Final (N)	4.568 ± 0.050 ^a^	3.328 ± 0.015 ^b^
pH	4.503 ± 0.014 ^a^	4.034 ± 0.001 ^b^
Aw	0.950 ± 0.000 ^a^	0.991 ± 0.001 ^b^
TTA (meq/g)	0.015 ± 0.000 ^a^	0.045 ± 0.001 ^b^
Lactic acid content (g/100 g)	0.030 ± 0.001 ^a^	0.334 ± 0.029 ^b^
Acetic acid content (g/100 g)	0.078 ± 0.005 ^a^	0.108 ± 0.005 ^b^
Lactic acid/Acetic acid ratio	0.387 ± 0.018 ^a^	3.056 ± 0.024 ^b^

Different superscript letters (a,b) in each row indicate significant differences between treatments according to Tukey’s HSD post hoc test (*p* < 0.05). Parameters include Initial and Texture_Final (N), Aw, TTA (meq/g), pH, Lactic acid content (g/100 g), Acetic Acid content (g/100 g), and the Lactic acid/Acetic acid.

**Table 3 foods-14-04236-t003:** Three-way ANOVA (F × T × A) summary of main and interaction effects on the physico-chemical and textural parameters of bread samples.

Parameters	F	T	A	F × T	F × A	T × A	F × T × A
Texture_Initial (N)	***	***	***	***	***	ns	ns
Texture_Final (N)	***	***	***	***	***	ns	ns
pH	***	***	***	***	***	ns	***
Aw	***	ns	ns	***	***	ns	ns
TTA (meq/g)	***	***	***	***	***	ns	**
Lactic acid content (g/100 g)	***	ns	ns	***	***	ns	***
Acetic acid content (g/100/g)	***	***	ns	***	***	ns	***
Lactic acid/Acetic acid ratio	***	***	ns	***	***	ns	***

Notes: F (Fermentation type), T (Temperature), A (Atmosphere). Statistical significance: *** *p* < 0.001, ** *p* < 0.01, ns = not significant.

**Table 4 foods-14-04236-t004:** Performance Metrics of the GLM for Binary Shelf-Life Classification.

Parameters	Value (%)	Note
Overall Accuracy ^1^	89.0 ± 1.5	Cross-validation result
Sensitivity (Recall) ^2^	91.5 ± 2.2	True positive rate
Specificity ^3^	86.5 ± 2.0	True negative rate
Precision ^4^	88.0 ± 1.2	Positive predictive value
F1-Score ^5^	89.7 ± 1.6	Harmonic mean of precision and recall
AUC-ROC ^6^	92.0 ± 1.2	Area under ROC curve

Notes: ^1^ proportion of correctly classified samples; ^2^ ability to identify acceptable samples; ^3^ ability to identify unacceptable samples; ^4^ probability that a sample predicted as acceptable is truly acceptable; ^5^ harmonic mean of precision and sensitivity; ^6^ overall discriminative ability of the model.

## Data Availability

The original contributions presented in this study are included in the article. Further inquiries can be directed to the corresponding author.

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
