# Peer review of "A Machine Learning Model to Reframe the Concept of Shelf-Life in Bakery Products: PDO Sourdough as a Technological Preservation Model"

_foods, 2025, doi:10.3390/foods14244236_

Round 1

Reviewer 1 Report

Comments and Suggestions for Authors

This work presents an interesting approach to understanding how a traditional sourdough with Protected Designation of Origin may act as a conditionally activated preservation mechanism, integrating classical statistical analysis with interpretable predictive modeling. However, while the topic is relevant, several aspects of the manuscript require major revision.

Major revisions:

> In Section 2.1 and Section 2.2, the authors refer to the Protected Designation of Origin procedure but do not provide operational details. 

> The authors use a fixed threshold for textural collapse, but it does not explain why that value was chosen or whether it reflects sensory acceptability or previous standards .

> The manuscript states that each measurement was taken from an independent sample, but it does not explain how variability between units was handled or whether any randomization was applied.

> I suggest clarifying the preprocessing steps used in Section 3.4. The score plot shows clear clustering, but the authors do not specify whether the variables were centered or scaled, how replicates were incorporated or whether potential batch effects were considered.

> The numerical thresholds in Equation 4 need clearer explanation, as the manuscript does not describe how the decision tree generated these cutoffs or whether their stability was evaluated.

Minor revisions:

- Several technical terms and abbreviations would benefit from clearer and more consistent introduction; for instance, titratable acidity, crumb firmness, and other descriptors appear without a uniform convention when first mentioned.

- Units for organic acids should be standardized, as the formatting in Table 2 is not fully consistent with the presentation in Figure 3.

- Some values use comma decimals, while others use period decimals (see Figure 4).

- Multivariate replicates should be documented, for example, in Section 3.4, the score plot indicates group separation, but the manuscript does not describe how replicates were distributed or whether the analysis accounted for batch effects.

- In Sections 2 and 3, there are inconsistent styles in reporting significance levels and statistical symbols. 

-

Author Response

REVIEWER 1

---------------------------------------

This work presents an interesting approach to understanding how a traditional sourdough with Protected Designation of Origin may act as a conditionally activated preservation mechanism, integrating classical statistical analysis with interpretable predictive modeling. However, while the topic is relevant, several aspects of the manuscript require major revision.

Major revisions:

> In Section 2.1 and Section 2.2, the authors refer to the Protected Designation of Origin procedure but do not provide operational details.

· We sincerely thank the Reviewer for this helpful observation. We have substantially expanded Sections 2.1 [94-96 rows] and 2.2 [106-118 rows] to include the complete operational workflow of the PDO sourdough process. These additions ensure full methodological transparency.

> The authors use a fixed threshold for textural collapse, but it does not explain why that value was chosen or whether it reflects sensory acceptability or previous standards .

· We appreciate the Reviewer’s remark which improve the understanding of the methodology of our work. A detailed justification for the structural-collapse threshold has been added to the 2.3.5 Texture analysis [223-229 rows] in Methods

section. Lastly, in 2.4 ML Model (GLM approach) section we expanded the definition of texturally acceptable [248-256 rows].

> The manuscript states that each measurement was taken from an independent sample, but it does not explain how variability between units was handled or whether any randomization was applied.

· We thank the Reviewer for pointing this out. We have added a dedicated explanation in 2.3 Physicochemical monitoring section [182-190 rows] to describe how variability was controlled through independent destructive replicates, randomized measurement order, rotation of loaves within oven batches, and randomized assignment to storage units. This clarification strengthens the methodological rigor.

> I suggest clarifying the preprocessing steps used in Section 3.4. The score plot shows clear clustering, but the authors do not specify whether the variables were centered or scaled, how replicates were incorporated or whether potential batch effects were considered.

· We appreciate the Reviewer’s suggestion. Section 3.4 now explicitly describes the preprocessing steps applied to the PCA: mean-centering, unit-variance scaling, treatment of replicates as independent observations, and inspection of potential batch effects in lines 447-449 and then, at the end of the paragraph in lines 476-479. No session-related clustering was detected.

> The numerical thresholds in Equation 4 need clearer explanation, as the manuscript does not describe how the decision tree generated these cutoffs or whether their stability was evaluated.

· Thank you for highlighting this issue. The accompanying text to Equation (4) has been revised to clarify that the thresholds were automatically generated by the decision tree during the model training. Moreover, their stability was assessed by refitting the model under multiple random seeds and resampling schemes. Minimal fluctuations was observed across fits [569-576 rows].

Minor revisions:

- Several technical terms and abbreviations would benefit from clearer and more consistent introduction; for instance, titratable acidity, crumb firmness, and other descriptors appear without a uniform convention when first mentioned.

- Units for organic acids should be standardized, as the formatting in Table 2 is not fully consistent with the presentation in Figure 3.

- Some values use comma decimals, while others use period decimals (see Figure 4).

- Multivariate replicates should be documented, for example, in Section 3.4, the score plot indicates group separation, but the manuscript does not describe how replicates were distributed or whether the analysis accounted for batch effects.

- In Sections 2 and 3, there are inconsistent styles in reporting significance levels and statistical symbols.

· We sincerely thank the Reviewer for drawing attention to these important stylistic inconsistencies. In the revised manuscript, we conducted a comprehensive, line-by-line review to ensure full harmonization of all abbreviations, measurement units, and statistical symbols. Specifically, we standardized the first appearance and subsequent use of all acronyms; we corrected units for moisture, organic acids, and water activity to follow SI and MDPI conventions; and we harmonized the reporting of statistical significance levels across the entire text, tables, and figure captions (p < 0.001, p < 0.01, p < 0.05, ns). This uniform formatting improves the clarity, readability, and professional presentation of the manuscript and aligns it fully with MDPI editorial guidelines.

Reviewer 2 Report

Comments and Suggestions for Authors

In the paper titled A Machine Learning Model to Reframe the Concept of Shelf-Life in Bakery Products: PDO Sourdough as a Technological Preservation Model,” Marianelli et al. explore how traditional sourdough, specifically the Tuscan Bread PDO starter, can act not only as a flavoring or leavening agent but also as a smart, condition-dependent preservation system.

The researchers tested breads made with either regular yeast or PDO sourdough under different storage temperatures and packaging atmospheres, then used machine learning (a Generalized Linear Model) alongside classical statistical methods to identify what truly drives shelf life. They found that sourdough can extend bread freshness up to 54 days, over three times longer than conventional bread, but only when the storage environment supports its natural acidification process. Rather than simply suppressing spoilage microbes, the sourdough changes the bread’s internal chemistry, lowering pH and increasing acidity in ways that stabilize texture and slow deterioration.

The study presents a novel approach, combining experimental baking trials with a predictive, data-driven model to show that sourdough preservation depends on reaching specific biochemical thresholds rather than being an automatic effect of using sourdough itself.

I have a couple of comments for the authors. Because shelf life naturally changes over time, using a method that examines when the bread fails, not just if it fails, could provide a clearer and more detailed picture. In this study, the authors simplified shelf life into a yes/no label (“acceptable” or “not acceptable”). While this makes the analysis easier, it also hides valuable information about how quickly bread quality declines under different conditions. For future studies, a survival analysis or time-to-failure approach could capture the exact point when the product becomes unacceptable and allow direct comparisons of spoilage rates between treatments. This would offer a deeper understanding of how sourdough and storage conditions influence the timing of deterioration, not just the final outcome.

The main concern in the results and discussion is that the authors draw relatively strong conclusions from a small and limited dataset. They clearly demonstrate that PDO sourdough extends shelf life and link this effect to changes in acidity and texture, but they present these findings as if the mechanism were fully established. With only 32 samples and a single bread type, the results should be described as promising but preliminary. The machine-learning model also appears highly accurate, but without external validation or a larger dataset, it remains uncertain whether the model would perform as well in other production settings.

Author Response

REVIEWER 2

---------------------------------------

In the paper titled “A Machine Learning Model to Reframe the Concept of Shelf-Life in Bakery Products: PDO Sourdough as a Technological Preservation Model,” Marianelli et al. explore how traditional sourdough, specifically the Tuscan Bread PDO starter, can act not only as a flavoring or leavening agent but also as a smart, condition-dependent preservation system.

The researchers tested breads made with either regular yeast or PDO sourdough under different storage temperatures and packaging atmospheres, then used machine learning (a Generalized Linear Model) alongside classical statistical methods to identify what truly drives shelf life. They found that sourdough can extend bread freshness up to 54

days, over three times longer than conventional bread, but only when the storage environment supports its natural acidification process. Rather than simply suppressing spoilage microbes, the sourdough changes the bread’s internal chemistry, lowering pH and increasing acidity in ways that stabilize texture and slow deterioration.

The study presents a novel approach, combining experimental baking trials with a predictive, data-driven model to show that sourdough preservation depends on reaching specific biochemical thresholds rather than being an automatic effect of using sourdough itself.

I have a couple of comments for the authors. Because shelf life naturally changes over time, using a method that examines when the bread fails, not just if it fails, could provide a clearer and more detailed picture. In this study, the authors simplified shelf life into a yes/no label (“acceptable” or “not acceptable”). While this makes the analysis easier, it also hides valuable information about how quickly bread quality declines under different conditions. For future studies, a survival analysis or time-to-failure approach could capture the exact point when the product becomes unacceptable and allow direct comparisons of spoilage rates between treatments. This would offer a deeper understanding of how sourdough and storage conditions influence the timing of deterioration, not just the final outcome.

The main concern in the results and discussion is that the authors draw relatively strong conclusions from a small and limited dataset. They clearly demonstrate that PDO sourdough extends shelf life and link this effect to changes in acidity and texture, but they present these findings as if the mechanism were fully established. With only 32 samples and a single bread type, the results should be described as promising but preliminary. The machine-learning model also appears highly accurate, but without external validation or a larger dataset, it remains uncertain whether the model would perform as well in other production settings.

· We appreciated the comments and sincerely we thank the Reviewer for the constructive and insightful feedback provided on the manuscript. In response, we have carefully reviewed the text to address all aspects raised in the comment. First, we clarified why survival analysis was not applicable in the present study [329-335 rows]. As detailed in Section 3.5, the experiment relied on destructive sampling, where each loaf was assessed only once and subsequently discarded. This design does not allow reconstruction of longitudinal deterioration trajectories, which are essential for time-to-event or hazard-based models. For this reason, shelf-life could only be modelled as a binary acceptability outcome

Second, we acknowledged the limitations associated with the relatively small dataset (n = 32). We have revised the Results and the Conclusions to present the findings as promising but preliminary, explicitly indicating that larger industrial batches will be required for external validation and for testing the generalizability of the observed patterns [602-603; 613-616; 619-627 rows].

Third, in line with the Reviewer’s concerns regarding the robustness of the GLM, we added a dedicated clarification in Section 3.5 describing that, although internal 10-fold cross-validation supports model consistency, an external validation step is still necessary to confirm the stability of decision boundaries and to strengthen predictive reliability across heterogeneous real-world production cycles [489-493 rows]. Together, these revisions improve methodological transparency, balance the interpretation of the GLM outcomes, and more clearly delineate the scope and limitations of the current work.

Reviewer 3 Report

Comments and Suggestions for Authors

The manuscript is good but it needs some revision before the manuscript can be accepted. 

@ The full-factorial design with 4 replicates per condition is whats written in the manuscript, but the manuscript doesn't include the information about how the sample size was determined statistically to ensure sufficient sensitivity for detecting PDO activation thresholds and interaction effects.

@ How sampling intervals influence the temporal resolution of GLM-based mediation analysis would be valuable. Consider adding reasoning for why the specific destructive sampling approach minimizes variability compared to repeated non-destructive measures.

@ Technical comments could address potential measurement error or calibration consistency across replicates (especially for pH, Aw, and TTA), which directly affect GLM inference.

@ I have a doubt on the implications of the small dataset (n = 32) on model stability and predictive uncertainty, especially given complete separation in categorical predictors. Can this be addressed in the manuscript. 

@ Consider adding recent relevant literature to strengthen the mechanistic and functional context of your bio-preservation system. In particular: Sun et al. (2025) DOI: 10.1021/acs.jafc.4c07078 and Li et al. (2026) DOI: 10.1016/j.foodhyd.2025.112017. These works may provide useful insight into polysaccharide‑mediated gut barrier effects and protein structural modifications under processing stress. I also recommend conducting a broader literature search for studies examining biochemical modulation (e.g., through polysaccharides, pH cycling, ultrasound) in food matrices, as they may reveal additional mechanistic parallels with your PDO preservation system.

Author Response

REVIEWER 3

---------------------------------------

The manuscript is good but it needs some revision before the manuscript can be accepted.

1. The full-factorial design with 4 replicates per condition is whats written in the manuscript, but the manuscript doesn't include the information about how the sample size was determined statistically to ensure sufficient sensitivity for detecting PDO activation thresholds and interaction effects.

· We thank the Reviewer for raising this important methodological point. We have now added a clear justification in the Methods (Section 2.4), explaining that the use of four destructive replicates per condition was determined based on the need to detect medium-to-large interaction effects in a full-factorial framework [304-307 rows]. This level of replication balances statistical sensitivity with the constraints of destructive sampling and aligns with common experimental practice in dough fermentation and bread quality studies. The added explanation clarifies the inferential adequacy of the chosen sample size.

2. How sampling intervals influence the temporal resolution of GLM-based mediation analysis would be valuable. Consider adding reasoning for why the specific destructive sampling approach minimizes variability compared to repeated non-destructive measures.

· We thank the Reviewer for this insightful methodological observation. In the revised manuscript, we have added a dedicated clarification in Section 2.2 (Breadmaking Process and Sampling Schedule) explaining the rationale behind the chosen sampling intervals and the use of a destructive sampling strategy [168-176 rows]. Specifically, we now state that sampling intervals were selected to increase temporal resolution in proximity to the expected PDO-mediated deterioration threshold, where physicochemical transitions occur more rapidly, while avoiding unnecessary early-stage measurements that would contribute limited information to the mediation model. We also clarify that destructive sampling minimizes measurement variability because each loaf is evaluated only once as an independent unit, avoiding noise introduced by repeated non-destructive assessments such as mechanical perturbation, moisture redistribution, cut-surface exposure, and structural fatigue. This approach ensures independence between observations and improves the stability and accuracy of GLM-based mediation inference.

3. Technical comments could address potential measurement error or calibration consistency across replicates (especially for pH, Aw, and TTA), which directly affect GLM inference.

· We thank the Reviewer for highlighting this important technical aspect. We have added a dedicated methodological note in Section 2.3 (Physicochemical monitoring) describing the calibration procedures performed before each analytical session, including standardized buffer calibration for pH, certified reference salts for Aw, and NaOH standardization for TTA. Additionally, we specify that duplicate readings were taken when appropriate to ensure analytical consistency [190-195 rows]. These clarifications directly address measurement reliability, which is essential for supporting the robustness of GLM-based inference.

4. I have a doubt on the implications of the small dataset (n = 32) on model stability and predictive uncertainty, especially given complete separation in categorical predictors. Can this be addressed in the manuscript.

· We appreciated this thoughtful concern and we remodulated discussing the limitation and the future work to be done for the validation of the model. As recommended, we have included a paragraph in Section 3.5 (GLM modeling) acknowledging that the limited dataset increases uncertainty around the stability of logistic decision boundaries and requires cautious interpretation [500-505 rows]. We also explain that complete separation arises from the deterministic structure of physicochemical activation rather than overfitting, but emphasize the need for external validation on larger, independent industrial datasets. This clarification improves transparency regarding predictive uncertainty and future validation needs.

5. Consider adding recent relevant literature to strengthen the mechanistic and functional context of your bio-preservation system. In particular: Sun et al. (2025) DOI: 10.1021/acs.jafc.4c07078 and Li et al. (2026) DOI: 10.1016/j.foodhyd.2025.112017. These works may provide useful insight into polysaccharide-mediated gut barrier effects and protein structural modifications under processing stress. I also recommend conducting a broader literature search for studies examining biochemical modulation (e.g., through polysaccharides, pH cycling, ultrasound) in food matrices, as they may reveal additional mechanistic parallels with your PDO preservation system.

· We sincerely thank the Reviewer for these valuable suggestions, and we added both to the reference list and in Conclusion section. We have now incorporated the two recommended references (Sun et al., 2025; Li et al., 2026), where we contextualize our findings within the broader field of biochemical modulation in food matrices [631-635]. These additions enrich the mechanistic depth of the manuscript while aligning it with current literature.

Round 2

Reviewer 3 Report

Comments and Suggestions for Authors

The comments are addressed. 

Author Response

We thank the referee for reviewing the paper